# Stability analysis of wheat lines with increased level of arabinoxylan

**Karolina Tremmel-Bede**[1], **Marietta Szentmiklóssy**[2], **Sándor Tömösközi**[2], **Kitti Török**[2], **Alison Lovegrove**[3], **Peter R. Shewry**[3], **László Láng**[1], **Zoltán Bedő**[1], **Gyula Vida**[1], **Marianna Rakszegi**[1]*

**1** Agricultural Institute, Centre for Agricultural Research, Martonvásár, Hungary, **2** Department of Applied Biotechnology and Food Science, Budapest University of Technology and Economics, Budapest, Hungary, **3** Department of Plant Science, Rothamsted Research, Harpenden, Hertfordshire, England, United Kingdom

* rakszegi.mariann@agrar.mta.hu

**Data Availability Statement:** All data files are available from REAL, Repository of the Hungarian Academy of Sciences at http://real.mtak.hu/id/

## Abstract

Plant breeders have long sought to develop lines that combine outstanding performance with high and stable quality in different environments. The high-arabinoxylan (AX) Chinese variety Yumai-34 was crossed with three Central European wheat varieties (Lupus, Mv-Mambo, Ukrainka) and 31 selected high-AX lines were compared for physical (hectolitre weight, thousand grain weight, flour yield), compositional (protein content, gluten content, pentosan) and processing quality traits (gluten index, Zeleny sedimentation, Farinograph parameters) in a three-year experiment (2013–2015) in the $F_7$-$F_9$ generations. The stability and heritability of different traits, including the relative effects of the genotype (G) and environment (E), were determined focusing on grain composition. The contents of total and water-soluble pentosans were significantly affected by G, E and G × E interactions, but the heritability of total (TOT)-pentosan was significantly lower (0.341) than that of water-extractable (WE)-pentosan (0.825). The main component of the pentosans, the amount and composition (arabinose: xylose ratio) of the arabinoxylan (AX), was primarily determined by the environment and, accordingly, the broader heritability of these parameters were 0.516 and 0.772. However, genotype significantly affected the amount of water-soluble arabinoxylan and its composition and thus the heritability of these traits was also significant (0.840 and 0.721). The genotypes exhibiting higher stability of content of TOT-pentosan also showed more stable contents of WE-pentosan. There was a positive correlation between the stability of contents of WE-pentosan and WE-AX, while the stability of the WE-AX content and AX composition were also strongly correlated. Water absorption was strongly genetically determined with a heritability of 0.829 with the genotype determining 38.67% of the total variance. Many lines were grouped in the GGE biplot, indicating that they did not significantly differ stability.

## Introduction

The development of high fiber cereal genotypes has received significant attention in the last decade. Tremmel-Bede et al. [1] developed high arabinoxylan (AX) lines and found a 0.5%

eprint/73492 (accession number 73492).
Tremmel-Bede K, Láng L, Török K, Tömösközi S,
Vida Gy, Shewry PR, Bedő Z, Rakszegi M.
Development and characterization of wheat lines
with increased levels of arabinoxylan. Euphytica
2017;213:291. https://doi.org/10.1007/s10681-
017-2066-2.

**Funding:** MR, KTB, GV, LL, and ZB received
support from the National Research, Development
and Innovation Office (NKFIH) of Hungary (project
No K112169) "Improvement of bioactive
composition and its effects." Partners of the same
research program ST, MS and KT received support
under project No K112179. AL and PRS received
grant-aided support from the Biotechnology and
Biological Sciences Research Council (BBSRC) of
the UK, with the work reported forming part of the
Designing Future Wheat Institute Strategic
Programme [BB/P016855/1], with additional
support from Innovate UK grant, L005654/1 "High
fibre wheat for healthier white bread." Cooperation
between all partners is supported by COST Action
18101 (European Cooperation in Science and
Technology) SOURDOMICS project with the title:
"Sourdough biotechnology network towards novel,
healthier and sustainable food and bioprocesses"
to MR. The funders had no role in study design,
data collection and analysis, decision to publish, or
preparation of the manuscript.

**Competing interests:** The authors have declared
that no competing interests exist.

**Abbreviations:** A/X, Arabinose to xylose ratio; AX,
Arabinoxylan; DF, Dietary fiber; E, Environment; G,
Genotype; GC, Gas Chromatography; GI, Gluten
index; LU, Lupus; MA, Mv-Mambo; TKW,
Thousand kernel weight; TOT, Total; TW, Test
weight; UK, Ukrainka; YU, Yumai-34; WA,
Farinograph water absorption; WE, Water-
extractable.

increase (24% if normal control is 100%) in the water-extractable (WE) AX content and 1% increase (42% relative) in the content of total (TOT) AX of the flour with an associated improvement in dough properties. Many lines showed an increase in the thousand-kernel weight, protein content, gluten content, Zeleny sedimentation and water absorption of the flour. The yield was competitive in three of the lines with the official control varieties, making them suitable for registration. The content and composition of AX were found to be genetically determined, especially for the WE fraction, but they were affected by agronomic conditions and the climate.

Twenty-six varieties were studied in four to six environments in an extensive study of the EU-FP6 Healthgrain program. They found that dietary fiber components are highly heritable in flour and bran. The heritability of TOT-AX, WE-AX in white flour and of β-glucan in wholemeal were 70%, 60% and 50% respectively [2–5]. Other studies also showed that AX content is highly heritable. Lempereur et al. [6] showed a 4.5 genotype/environment (G/E) ratio for TOT-AX and 4.9 for WE-AX by analysing wholemeals originating from durum seeds grown under four agronomic regimes. Finnie et al. [7] indicate that the variation in arabinoxylan content is primarily influenced by cultivar and secondarily influenced by environment in analyses of seven spring wheat lines grown in 10 environments and 20 winter wheat lines grown in 12 environments. Hong et al. [8] assessed 60% higher effect of G then E for WE-pentosans and 140% greater effects for TOT-pentosans by analysing 18 wheat wholemeal samples grown at two sites. Martinant et al. [9] determined the broad sense heritability (genotypic variance/phenotypic variance) to be 0.75 for WE-AX of flour and 0.80 for the viscosity of water extracts of flour by comparing 19 varieties grown at three growing sites. Similarly, Dornez et al. [10] calculated broad sense heritabilities of 0.53 for TOT-AX and 0.96 for WE-AX in wholemeal of 14 cultivars grown for three years. By contrast, Li et al. [11] and Török et al. [12] reported much stronger effects of environment than of genotype for both TOT-AX and WE-AX, based on comparisons of 25 winter and 25 spring wheat genotypes at three growing sites and of 41 genotypes grown for three years, respectively. The results from these studies therefore suggest that WE-AX is a good target for breeding, but the results were less consistent for TOT-AX [13].

Other studies have tried to identify correlations between grain composition and individual environmental factors. Gebruers et al. [5] found strong negative correlations between WE-AX and the average daily temperature while the correlation was positive with total precipitation both in flour and in bran. However, hey could not show significant correlations between TOT-AX and weather conditions. Coles et al. [14] identified positive relations between drought stress and the TOT-AX content, but other studies have shown that the impact of drought on AX content is also affected by the strength and timing of drought [8, 15, 16]. The development of varieties or breeding lines which give stable performance and/or quality across a wide range of environments has long been a target of breeders and methods have been developed to determine the stability of genotypes [17, 18]. The development of stable lines is especially important in areas with diverse conditions or if strong environmental fluctuations occur over the years in the same ecogeographical region.

Most authors have studied the stability of the yield performance of wheat genotypes under different environmental conditions. For example, Abraha et al. [19] in Ethiopia, Singh et al. [20] in India and George and Lundy [21] in California studied the performance of the genotypes under rainfall conditions, while Matlala et al. [22] studied the stability of the dryland wheat genotypes in South Africa. Thungo et al. [23] compared the effects of heat and drought stress on wheat grown in the glasshouse and the field with rainout shelters in order to identify genotypes with stable high performance. Similarly, Karaman et al. [24] compared irrigated and rainfed conditions, while Beres et al. [25] considered both moisture and N fertilitisation in

their studies in Canada. Other studies have also determined effects on grain morphology and/ or yield components [26, 27], but less studies are available on the stability of grain compositon and quality. Both grain yield and some quality traits (thousand kernel weight, test weight, protein, starch, gluten, Zeleny, Alveograph) were determined to identify stable wheat genotypes in Turkey [28, 29, 30], and in India [31], while Silva et al. [32] also considered the effect of the sowing dates on the stability of these traits in Brazil. Amiri et al. [33] and Tremmel-Bede et al. [34] have determined the stability of compositional traits focusing on proteins and fibers. The former Amiri et al. [33] compared old and new varieties, showing that new varieties had higher grain yields and gluten indises and lower contents of protein and fiber. Tremmel-Bede et al. [34] studied the stability of diverse populations and the effect of field management practices, showing that diversity within the population have stabilizing effects on grain composition which were more highly expressed under low-input systems.

The present study therefore aimed to determine the stability of the breeding lines with high arabinoxylan content and the effects of genotype-environment interactions on the compositions and properties of three generations grown over consecutive years.

## Materials and methods

### Plant materials

In order to combine high AX content with adaptation to European conditions and good breadmaking quality, crosses were made between the Chinese wheat cultivar Yumai-34 (released in 1988) and three wheat cultivars (Lupus, Mv-Mambo, and Ukrainka) [1].

Yumai 34 was identified in the EU FP6 HEALTHGRAIN project as having high contents of both TOT-AX and WE-AX in white flour [35] while the three other varieties had good adaptation traits under European environmental conditions: optimal plant growth development, good abiotic stress resistance (winter hardiness, sprouting resistance) and high productivity. Spikes from the $F_2$ segregating populations were planted in $F_3$ headrows. Selection of the plants were carried out then for agronomic traits and high contents of protein and water-extractable (WE)- pentosan (as a measure of AX) in flour in each generation. Finally 31 lines were selected (12 Lupus/Yumai-34, three Mv-Mambo/Yumai-34, and 16 Ukrainka/Yumai-34) and analysed in the $F_7$, $F_8$ and $F_9$ generations (2013–2015), all being over 97% homozygous. The parental wheat cultivars (Yumai-34, Lupus, Mv-Mambo, and Ukrainka) were used as controls.

### Growing conditions

The lines were grown at the Agricultural Institute, Centre for Agricultural Research, Martonvásár (latitude, 47° 21' N; longitude, 18° 49' E; altitude, 150 m) in 2013, 2014, and 2015. The plots were 2.5 m long with six rows spaced at 20 cm. In the third year of the experiment the performance of the lines was compared in a field experiment with 6 m long plots organised in a randomised complete block design of three replicates. The soil was a chernozem with a loam texture and pH 7.25. The previous crops were oil radish (2012/2013, 2013/2014), and phacelia (2014/2015). The plots were treated with herbicide (4 L/ha U-46 D-fluid SL containing 500 g/L 2-methyl-4-chlorophenoxyacetic acid, 40 g/ha Granstar 50 SX containing 50% tribenuron methyl), insecticide (0.2 L/ha Karate Zeon 5CS containing 50 g/L λ-cihalotrin), and fungicide (first: 1 L/ha Amistar Extra containing 200 g/L azoxistrobin and 80 g/L ciprokonazol, second: 1 L/ha Cherokee containing 50 g/L ciprokonazol, 62 g/L propiconazol and 375 g/L cloretalonil) each year. The growing conditions in the years of the experiment: 2013/2014 had mild temperatures with high precipitation before harvest, while 2012/2013 and 2014/2015 had a very hot and dry summer with a very low minimum temperature in winter in 2014/2015. (S1 and S2

Tables, OMSZ homepage: https://www.met.hu/eghajlat/magyarorszag_eghajlata/eghajlati_visszatekinto/elmult_evek_idojarasa/)

### Physical grain properties

The test weight (TW, g seed/100 liter) was measured by Foss Infratec 1241 (FOSS Tecator, Sweden) instrument according to the standard method: MSZ EN ISO 7971–3:2019 [36]. Thousand kernel weight (TKW, g/1000 kernels) was measured by Marvin Seed Analyser (Marvitech Gmbh, Germany) according tho the standard method: MSZ EN ISO 520:2011 [37]. Measurements were carried out with two technical replications.

### Grain composition

Seeds of each sample (500-500g) were conditioned to 15.5% moisture content and then Chopin CD1 Laboratory Mill was used to produce white flour. Kjeldahl method consistent with ICC method 105/2 [38] was used to measure crude protein content by FOSS Kjeltec 1035 Analyzer while Perten Glutomatic 2200 (ICC 137/1) [39] was used to measure gluten content. In order to select lines with high AX, the contents of TOT- and WE-pentosans were determined using a colorimetric method of Douglas [40]. AX was also measured in each generation at Budapest University of Technology and Economics (BUTE) by GC of monosaccharides as described by Gebruers et al. [41]. Measurements were carried out on two parallel samples with two technical replications.

### Dough properties

Dough properties (water absorption, dough development time, dough stability and dough softening) were measured by Brabender Farinograph according to the ICC 115/1 standard [42] method. Gluten index was calculated according to the ICC 155 standard [43].

### Statistical analyses

GGE biplot analysis was carried out using GenStat 17.0 software (VSN International Ltd., Hemel Hemstead, UK), while the discriminant analysis and linear mixed model analysis were carried out using SPSS 16.0 software (SPSS Inc., Chicago, IL, USA).

GGE biplot illustrated the genotype plus genotype-by-environment variation using scores from a principal component analysis, but removing the environmental effects. The Ranking biplot (average-environment coordination (AEC) view of the GGE biplot) can be used to examine the performances of all genotypes within a specific environment. In the plot, the best performing and most stable genotypes are those whose projections onto the biplot axis are closest to the environment. The single-arrowed line is the AEC abscissa pointing to higher mean values for a given trait. The AEC ordinate points to greater variability (poorer stability) in both directions [18]. The coefficient of correlation (CV) was also calculated with Microsoft Excel as a ratio of standard deviation and mean value multiplied by one hundred.

Linear Mixed Model analysis (using the restricted likelihood algorithm, REML) was carried out using SPSS 16.0 software (SPSS Inc., Chicago, IL, USA) based on Virk et al. [44]. A total of three years were regarded as different environments (E) for all the genotypes (G). The repeatability, genotypic variance, and variance of the G × E interaction were evaluated for each trait. Repeatability (broad-sense heritability) was calculated as the ratio of genotypic to phenotypic variance.

## Results

### Effect of the genotype and the environment

The effects of the genotype (G) and environment (E) and the interactions between these two factors (GxE) on the physical, compositional and breadmaking quality traits of wheat breeding lines containing increased level of TOT- and WE-AX was studied (Table 1). In this study, the three environments are different years of growth.

The test weight and the thousand kernel weight were both significantly affected by the environment (E.), while the genotype (G) and G x E interactions also affected the thousand kernel weight (TKW). This is reflected in the broad-sense heritability of TKW which was 0.786 (Table 1). The flour yield was strongly affected by the environment which was in agreeement with the low heritability value. The contents of protein and gluten and all the breadmaking quality parameters which depend on protein content and composition were strongly affected by the G resulting high broad-sence heritability values for protein content (0.851), gluten content (0.845) and gluten index (0.861). Water absorption is strongly affected by the degree of starch damage which occurs on milling, and is therefore affected by grain hardness and other factors, was also highly heritable (0.829). The Faringraph parameters (quality number, dough stability, softening and development time) showed lower heritability values (0.502–0.613) indicating more significant effects of E and G x E. Starch content was significantly determined by G, E and G x E with a heritability of 0.828. Arabinoxylan is the major dietary fibre component in wheat flour and the water-extractable (WE) and total (TOT) AX fractions were determined directly as monosaccharides using gas chromatography and as pentosans using a colorimetric method. The TOT fractions from both analyses showed similar strong effects of E with weaker effects of G, with heritabilities of 0.341 and 0.516. The WE fractions showed stronger effects of

**Table 1. Effect of the genotype (G), environment (E.), and their interaction (GxE) on the physical, compositional and breadmaking quality traits of 31 breeding lines using linear mixed model analysis and the heritability values ($h^2$) of the individual traits.**

| Properties | | G | E | GxE | $h^2$ |
|---|---|---|---|---|---|
| Test weight | kg/hl | n.s. | *** | n.s. | 0.322 |
| Thousand kernel weight | g/1000 seed | *** | *** | * | 0.786 |
| Flour yield | % | n.s. | *** | * | 0.202 |
| Protein content | % | *** | n.s. | n.s. | 0.851 |
| Starch | % | *** | *** | ** | 0.828 |
| TOT-pentosan | mg/g | ** | *** | *** | 0.341 |
| TOT-AX | % | n.s. | *** | ** | 0.516 |
| TOT-AX A/X | | n.s. | * | ** | 0.372 |
| WE-pentosan | mg/g | ** | *** | * | 0.825 |
| WE-AX | % | ** | *** | * | 0.840 |
| WE-AX A/X | | * | *** | * | 0.721 |
| Gluten content | % | *** | n.s. | ** | 0.845 |
| Gluten index | | *** | n.s. | ** | 0.861 |
| Faringraph quality number | | *** | * | * | 0.688 |
| Dough stability | min | ** | * | ** | 0.502 |
| Dough softening (12) | FU | ** | *** | ** | 0.555 |
| Dough development time | min | *** | * | ** | 0.613 |
| Water absorption of the flour | % | *** | *** | n.s. | 0.829 |

n.s.–not significant, *,**,***—significant at 0.05, 0.01 and 0.001 probability level.

G and E, with heritabilities above 0.8. G had a significant effect on the A/X ratio of the WE-AX fraction with a heritability of 0.721, but little effect on the A/X ratio of TOT-AX, which had low heritability (0.372).

The variance components of the different properties were also determined and are presented in Fig 1. As the main breeding target was increased content of dietary fiber, the discussion will focus on components of this fraction: TOT-pentosan, WE-pentosan, WE-AX and the

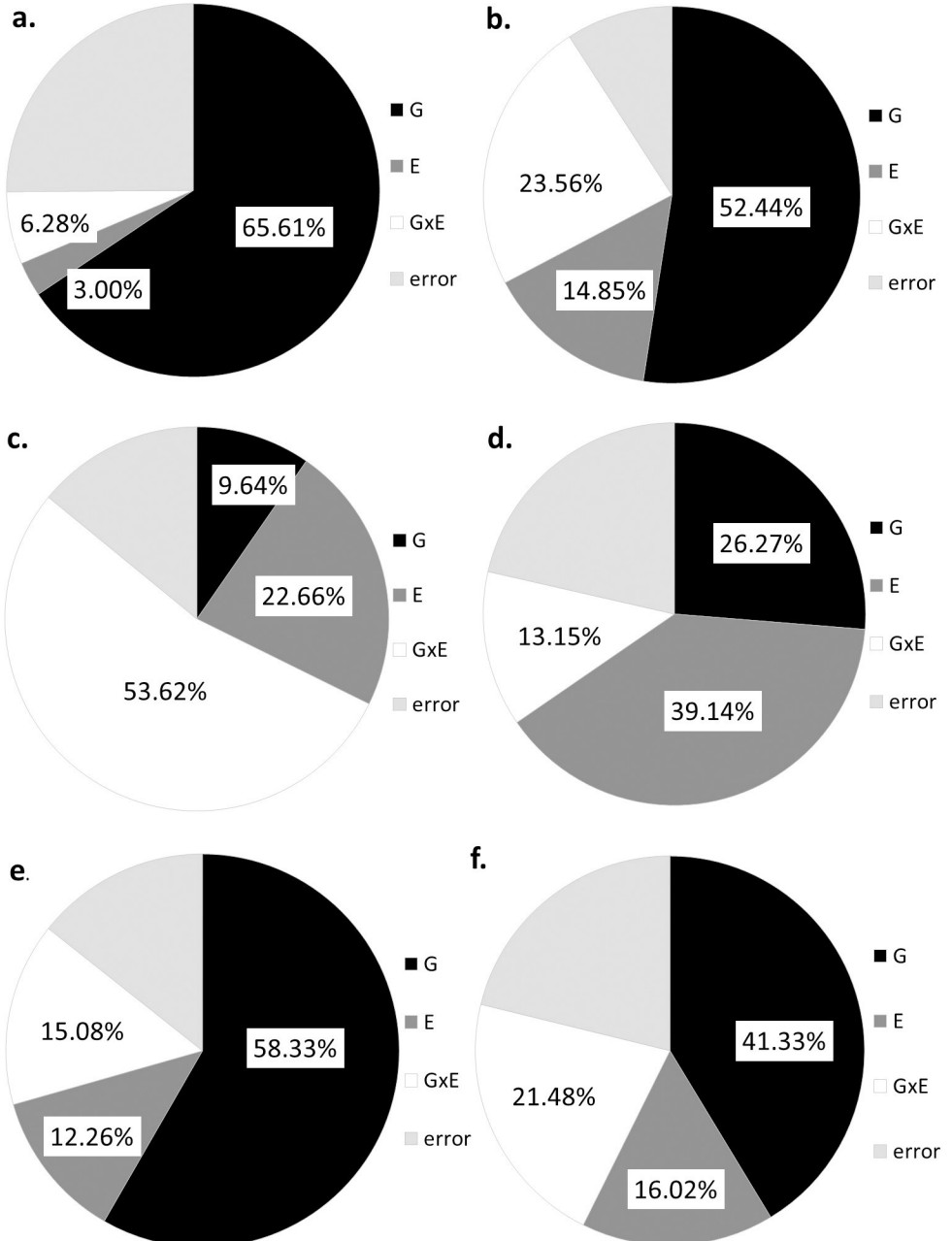

**Fig 1. Variance components determined for some compositional traits of the 31 breeding lines and their controls.** (a. protein content, b. starch content, c. total pentosan content (TOT pentosan), d. water-extractable pentosan content (WE-pentosan), e. water-extractable arabinoxylan content (WE-AX), f. arabinose/xylose ratio in WE-AX (A/X ratio) (2013–2015).

WE-AX A/X. The contents of protein and starch will also be discussed as basic compositional traits but TOT-AX and its composition are not considered because there was no effects of G and the error term was high.

The contents of protein and the starch were very highly determined by G, which accounted for 65.61% and 52.44% of the total variance, respectively (Fig 1). E determined relatively low proportions of the total variance (3.00% and 14.85%, respectively) although the G x E effect was relatively high for starch (23.56%). The contribution of G to total variance was low for TOT-pentosan (9.64%) while the effects of E (22.66%), and particularly G x E (53.62%) was much higher for TOT-pentosan. G determined 26.27% of the total variance of WE-pentosan and the contribution was even higher when the main component of pentosan, WE-AX, was determined (58.33%). Environment determined 39.14% and 12.26% of the total variance for WE-pentosan and WE-AX, respectively. The composition of WE-AX (A/X ratio) was also highly determined by the genotype (41.33%), but the environment and G x E also contributed to the total variance (by 16.02 and 21.48%, respectively).

## Stability of grain composition

GGE ranking biplot analysis was carried out in order to study the stability of the compositional properties of the breeding lines, focusing on protein, starch, WE-AX/pentosan and their compositions as the selection during wheat breeding was carried out based on protein and fiber content.

The stable genotypes were close to the AEC abscissa, where the arrow also showed the direction of the better performance (Fig 2). Thus, those lines which were located on the right side of the figures close to the AEC abscissa had the best performance and stability. Some lines had stable high values only in specific environments (years), those are located close to the sign of the given year. The other axis shows the grand means of the environments and the genotypes.

The lines with the highest protein contents were LU/YU_8,9,10,12, with LU/YU_9 also have the most stable content, while LU/YU_11 had the highest starch content. Those genotypes which were the best performing for protein content were the worst performing for starch content and vice versa. Yumai-34 had the highest protein content in 2014.

The three low AX parents (Ukrainka, Lupus and Mv-Mambo) all had low contents of TOT- and WE-pentosan, with Ukrainka having the lowest contents, while the high AX parent Yumai 34 had among the highest contents of these fractions. The genotype with the highest content of TOT-pentosan was UK/YU_1, while LU/YU_1 had the highest content of WE-pentosan. The distribution of the genotypes on WE-AX figure (Fig 2) was similar to that of WE-pentosan, with Yumai-34 and UK/YU_5 having the best WE-AX with highest stability.

Two of the parents, Lupus and Ukrainka showed low stability for composition of the WE-AX (the arabinose to xylose ratio) while UK/YU_15 showed high stability.

In general, the genotypes with high protein contents also had high TOT-pentosans but this was not related to lower kernel size and lower starch content of the seed [1].

In this study, the relationships between the stability of the measured parameters were determined by correlation of their coefficients of variation (CV) (Table 2). The lines that had more stable thousand kernel weights also had more stable starch contents, while a more stable content of starch was also associated with a more stable protein content. The strongest correlation was between the stability of the contents of TOT and WE-pentosans, with additional correlations between the stability of the contents of WE-pentosan and WE-AX, and between the stability of WE-AX content and its composition (A/X ratio). In contrast, the water uptake stability of the flour showed a negative association with the stability of the WE-AX content and composition (A/X).

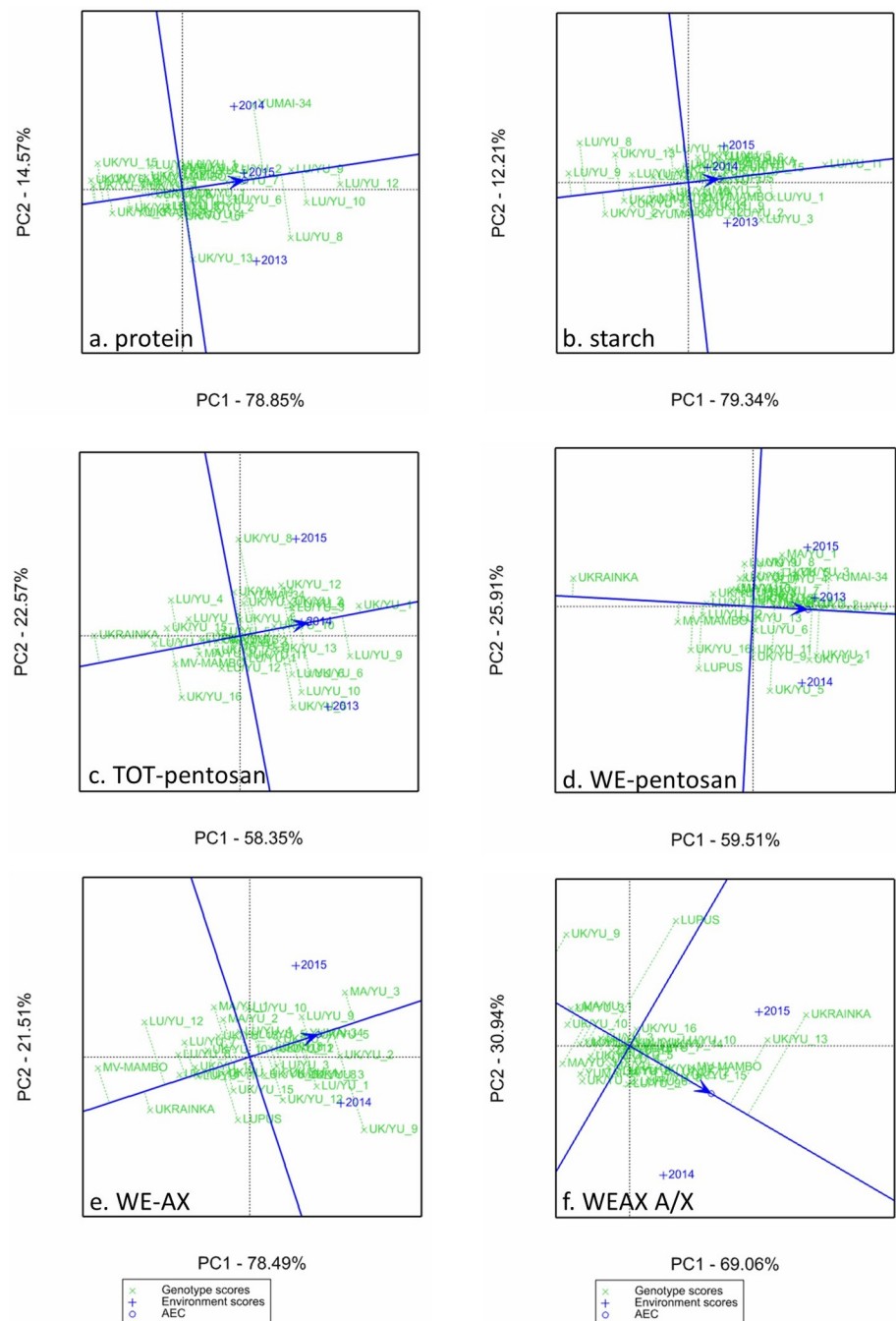

**Fig 2. Stability of the 31 lines and their controls based on their compositional qualtity traits.** (such as: protein content, starch content, total pentosan content (TOT-pentosan), water extractable pentosan content (WE-pentosan), water extractable arabinoxilan (WE-AX), arabinose/xylose ratio in WE-AX (A/X ratio) using GGE Biplot analysis (2013–2015).

Although the CV values indicate whether a given trait of a genotype is stable under various environmental conditions, it does not show whether the value represents a consistently high or low performance. This information can be obtained from the GGE biplot analysis, in which the genotypes that fall in the right arrow of the AEC abscissa are those that perform best for a

**Table 2. Correlation of the coefficient of variation values (CV).**

| | TKW | Protein | Starch | TOT-pentosan | WE-pentosan | WE-AX | WE-AX A/X | Wabs |
|---|---|---|---|---|---|---|---|---|
| TKW | 1.000 | | | | | | | |
| Protein | -0.007 | 1.000 | | | | | | |
| Starch | 0.376 | 0.490 | 1.000 | | | | | |
| TOT-pentosan | 0.107 | -0.081 | -0.136 | 1.000 | | | | |
| WE-pentosan | 0.084 | -0.269 | -0.040 | 0.671 | 1.000 | | | |
| WE-AX | 0.315 | -0.151 | 0.321 | 0.063 | 0.393 | 1.000 | | |
| WE-AX A/X | -0.206 | -0.056 | -0.061 | 0.116 | 0.409 | 0.511 | 1.000 | |
| Wabs | -0.081 | -0.156 | -0.430 | -0.079 | -0.156 | -0.343 | -0.385 | 1.000 |

$r_{5\%} = 0.3246$, $r_{1\%} = 0.4182$, $r_{0.1\%} = 0.5189$ –critical values of the correlation coefficients at n = 35.

A/X- arabinose to xylose ratio, TKW- thousand kernel weight, TOT- total, Wabs- water absorption of the flour, WE- water extractable.

given trait. This plot also allows the identification of the most stable lines with good performance as these fall closest to the AEC abscissa.

These lines were LU/YU_9 and LU/YU_11 for contents of protein and starch, UK/YU_1 and LU/YU_1 for TOT and WE-pentosan, UK/YU_5 for WE-AX and UK/YU_15 for WE-AX composition (A/X). Hence, we did not identify any genotypes that showed high stability for several traits. It should be noted, however, that several lines are grouped in the GGE biplot, which indicates that there is no significant difference in the stability of these lines. For each trait, only a few lines and varieties could be identified which differed significantly in their stability from the average of the varieties and environments.

## Discussion

Wheat protein content is known to be a multigenic trait which is highly influenced by the environment and therefore has very low heritability (0.2–0.3) [9, 45, 46]. The heritability of starch content is even lower, as the amount is influenced by many factors. In this study, analysis of 31 multiply selected genotypes with high contents of AX revealed that the contents of both protein and starch were strongly genetically determined, with high heritabilities (0.851 and 0.828, respectively). In addition, the starch content also showed very significant year (E) and G x E effects (Table 1). The high effect of the genotype on the protein and starch content in this study might be resulted by fact that the studied lines went through several cycles of selection during the different generations and those lines were selected which not only have high AX, but also have high protein content and/or quality.

All three factors (G, E, G x E) significantly influenced the contents of TOT and WE-pentosan, but the heritability of TOT-pentosan was significantly lower (0.341) than that of WE-pentosan (0.825). Similarly, when the major component of pentosan, arabinoxylan, was determined by gas chromatography the G had significant effects on both the amount and composition (A/X) of WE-AX with high heritabilities (0.840 and 0.721) while the heritabilities of the amount and composition of TOT-AX were lower (0.516 and 0.372) with stronger effects of E.

Our results are in agreement with that of Dornez et al. [10] who found lower heritability values for TOT-AX (0.53) than for WE-AX (0.96), but several other studies have found higher heritability values for TOT-AX than for WE-AX [5, 8].

Our results also disagree with some studies showing that the E has more significant effect on WE-AX than the G [11, 12].

Hence, most published studies support our finding that the heritability of WE-AX is high and thus a suitable target for breeding. However, the data for heritability of TOT-AX content vary greatly between studies. This may be due to differences in the methods used and/or the genotypes studied.

In addition to genetic factors, environmental factors may also have significant impacts on grain composition. These factors include the weather (precipitation, temperature, etc.), the soil composition and properties, agronomy (tilling, manure, previous crops), and the application of agrochemicals (pesticides and fertilizers). The most extreme environmental stresses are drought and heat, which are being experienced more frequently in Hungary.

Drought is one of the most damaging environmental stresses that can affect plants in southern Europe. Depending on its frequency and duration, wheat yields can be reduced by up to 50% [47, 48].

Drought stress is particularly harmful during flowering and the grain filling period, resulting in the reduction in the two main yield components: grain number and grain size [48]. However, drought stress also has a significant effect on the chemical composition of the grain, including the amount and composition of the storage proteins (gliadins, glutenins) and the fiber components (AX, β-glucan) [16, 49, 50]. In general, drought stress decreases the carbohydrate content of the grain (including sucrose and starch) [51, 52] and increases the protein content [53]. Under natural conditions, drought stress is usually associated with heat stress and the two factors act synergistically to increase the fiber content of wheat [5], including AX [16]. However, the overall impact is highly dependent on the severity and duration of the drought and the interaction with other environmental stresses. These environmental stresses also influence processing properties. For example, according to Li et al. [54], drought increases the dough strength while decreasing the bread volume. In our experiment, heat and drought stress was observed in the 2012/2013 and 2014/2015 seasons (S2 Table), with 36 and 37˚C maximum temperature and low amount of precipitation in the last 100 days (102.6 mm, 115.5 mm) of the plant development, which may have affected the properties of the grain grown in this year.

Correlation analysis was therefore carried out between the yearly average quality data and the weather conditions (S3 Table). The contents of protein, gluten and pentosan were correlated with the cumulative precipitation, the absolute minimum temperature and the number of days with higher temperature then 30˚C, showing the high dependency of compositional traits on drought and heat. At the same time the gluten index, the Zeleny sedimentation and the water absorption were correlated with the absolute minimum temperature in the last 100 days of plant development and the number of days with temperatures above 25˚C. Test weight, thousand kernel weight and dough softening were highly correlated with the absolute maximum temperature, the cumulative precipitation, the mean temperature and the absolute maximum temperature of the last 100 days and the number of low temperature days.

Earlier it was found, that the most important quality trait significantly correlated to the TOT-AX content is the water absorption [1]. Beyond that, only the starch content showed negative while the gluten content showed positive correlation to TOT-AX, but this last result might be related to the seed size. Consequently the water absorption will be discussed further on.

Water absorption was highly determined by G with a heritability of 0.829 and the G determining 38.67% of the total variance (Table 1 and Fig 1). The LU / YU_8,9,10 lines had the highest water absorptions and the highest contents of protein and pentosan. Although pentosans only comprise 2–3% of flour, they are able to absorb up to ten times their own weight of water, representing a quarter of the moisture content of the dough [55, 56]. Rakszegi et al. [57] reported that, in addition to the protein content and the level of starch damage, the total arabinoxylan content and the soluble protein content should also be considered when estimating the water absorption. Furthermore, Tremmel-Bede et al. [1] found a strong positive

correlation between total arabinoxylan and water absorption, although soluble small and medium sized arabinoxylans have a greater effect on water uptake [58].

Fiber (bran) interacts with the gluten-starch matrix during kneading [59] and competition for binding water occurs between these components [60, 61]. The high water-binding capacity of the fiber components results in higher water absorption determined by the Farinograph, which is advantageous to some extent for the processing industry [62]. However, at the same time the gelling properties of starch may also change [63]. Water uptake is influenced by several properties of the fiber, such as molecular weight and molecular size [61] and solubility [64]. For example, large fiber molecules absorb water much more slowly than small fiber molecules. Due to the increased water uptake of flour, the crispness of the crust is reduced, which is an undesirable property of bread. Increasing the fiber content also decreases dough extensibility while the dough development time increases [65]. Small amounts of non-starch polysaccharides, such as AX, can significantly increase dough resistance to kneading and the stability [66]. Moderate increases in these polymers may therefore be suitable for the enhancement of weaker flours [60].

## Conclusions

Analysis of lines with high contents of WE-fibre (determined as pentosan and AX) showed that these were suitable materials for breeding, as the contents of these components were highly heritable with genotype determining 26.27 and 58.33% of the total variance, respectively. They also showed good and stable processing properties, including water absorption, indicating the good quality and improved health benefits can be combined in commercial wheat breeding.

## Supporting information

**S1 Table. Main growing conditions in different years of the experiment.** (Martonvásár, 2013–2015 harvest years).
(DOCX)

**S2 Table. Weather conditions in three seasons.** (Martonvásár, 2013–2015 harvest years).
(DOCX)

**S3 Table. Correlations between the contents and quality and weather conditions in 31 wheat lines grown in three years.**
(DOCX)

## Author Contributions

**Conceptualization:** Sándor Tömösközi, Peter R. Shewry, Zoltán Bedő, Marianna Rakszegi.

**Data curation:** Sándor Tömösközi, Marianna Rakszegi.

**Formal analysis:** Kitti Török, Alison Lovegrove, Peter R. Shewry, László Láng.

**Funding acquisition:** Sándor Tömösközi, Alison Lovegrove, Peter R. Shewry, Zoltán Bedő, Marianna Rakszegi.

**Investigation:** Karolina Tremmel-Bede, Marietta Szentmiklóssy, Sándor Tömösközi, Alison Lovegrove, Zoltán Bedő, Marianna Rakszegi.

**Methodology:** Marietta Szentmiklóssy, Kitti Török, Alison Lovegrove, Zoltán Bedő, Gyula Vida.

**Project administration:** Karolina Tremmel-Bede.

**Resources:** László Láng, Zoltán Bedő, Gyula Vida, Marianna Rakszegi.

**Software:** Gyula Vida.

**Supervision:** Sándor Tömösközi, Alison Lovegrove, Peter R. Shewry, Zoltán Bedő, Marianna Rakszegi.

**Validation:** Marietta Szentmiklóssy, Sándor Tömösközi, Alison Lovegrove, Peter R. Shewry, Marianna Rakszegi.

**Visualization:** Karolina Tremmel-Bede, Kitti Török.

**Writing – original draft:** Karolina Tremmel-Bede, Marietta Szentmiklóssy, Sándor Tömösközi, Alison Lovegrove, Peter R. Shewry, Marianna Rakszegi.

**Writing – review & editing:** Peter R. Shewry, Marianna Rakszegi.

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
