## [Decision Letter · Decision Letter 0]

30 Mar 2020

PONE-D-20-06968

Stability analysis of wheat lines with increased level of arabinoxylan

PLOS ONE

Dear PhD Rakszegi,

Thank you for submitting your manuscript to PLOS ONE. After careful consideration, we feel that it has merit but does not fully meet PLOS ONE’s publication criteria as it currently stands. Therefore, we invite you to submit a revised version of the manuscript that addresses the points raised during the review process.

We would appreciate receiving your revised manuscript by May 14 2020 11:59PM. To enhance the reproducibility of your results, we recommend that if applicable you deposit your laboratory protocols in protocols.io, where a protocol can be assigned its own identifier (DOI) such that it can be cited independently in the future. For instructions see: http://journals.plos.org/plosone/s/submission-guidelines#loc-laboratory-protocols

We look forward to receiving your revised manuscript.

Kind regards,

Aimin Zhang, Ph.D.

Academic Editor

PLOS ONE

Journal Requirements:

https://link.springer.com/article/10.1007%2Fs10681-017-2066-2

https://www.sciencedirect.com/science/article/pii/B9780857090386500073?via%3Dihub

In your revision ensure you cite all your sources (including your own works), and quote or rephrase any duplicated text outside the methods section. Further consideration is dependent on these concerns being addressed.

3. Please upload a copy of Figure 4, to which you refer in your text on page 18. If the figure is no longer to be included as part of the submission please remove all reference to it within the text.

Reviewers' comments:

Reviewer's Responses to Questions

**Comments to the Author**

1. Is the manuscript technically sound, and do the data support the conclusions?

Reviewer #1: Yes

Reviewer #2: Yes

2. Has the statistical analysis been performed appropriately and rigorously? 

Reviewer #1: Yes

Reviewer #2: Yes

3. Have the authors made all data underlying the findings in their manuscript fully available?

Reviewer #1: Yes

Reviewer #2: No

4. Is the manuscript presented in an intelligible fashion and written in standard English?

Reviewer #1: Yes

Reviewer #2: Yes

5. Review Comments to the Author

Reviewer #1: The manuscript describes a study about a range of grain physical, chemical and quality traits of a suite of breeding lines via classic quantitative genetics approach. Based on my knowledge, the article is well written and contain some interesting discoveries. It involves a large amount of work which are professional designed and conducted. The manuscript is written in a clear and logical way, which is easy to understand. The conclusion well follows the results. In my view, this manuscript can be accepted for publishing on its current stand.

Reviewer #2: The paper by Marianna et al. analyzed the stability of wheat lines with increased level of arabinoxylan in three years experiment. The author found that the contents of WE-fibre were highly heritable and wheat lines with high contents of WE-fibre also have good and stable processing properties. With no doubt, the research can arouse interests for the readers who engage in breeding wheat cultivars with the good quality and improved health benefits. However, the paper should add more detailed information in order to increase its theoretical component. The manuscript also has some grammar errors and inaccuracies should be revised. Some detailed suggestion for author as below:

1. Please describe the agricultural traits and quality traits of the 31 improved lines and 4 parents。

2. The contents of protein and starch are low heritability in many reports. However, this research shows the heritability of the contents of protein and starch is very high ((0.851 and 0.828, respectively). Why the results of this research are different from other reports, please discuss more about this.

3. In the methods part, please supply the methods for the test weight and thousand kernel weight.

4. The significant marker in Table1 should be * instead of X, please revise it .

5. Line 89, there should be a full stop(.) after the ‘respectively’.

6. Line 347, ‘heritability values for TOT-AX then for WE-AX ‘ should be ‘heritability values for TOT-AX than for WE-AX ‘.

7. Line 349, ‘WE-AX then the G’ should be ‘WE-AX than the G’.

6. PLOS authors have the option to publish the peer review history of their article (what does this mean?). If published, this will include your full peer review and any attached files.

Reviewer #1: No

Reviewer #2: No

---

## [Author Response · Author response to Decision Letter 0]

20 Apr 2020

Response to reviewers

Thank you for the reviewers for their work and valuable additions to the manuscript quality. Here are my answers to their comments bellow:

Reviewer #1: 

The manuscript describes a study about a range of grain physical, chemical and quality traits of a suite of breeding lines via classic quantitative genetics approach. Based on my knowledge, the article is well written and contain some interesting discoveries. It involves a large amount of work which are professional designed and conducted. The manuscript is written in a clear and logical way, which is easy to understand. The conclusion well follows the results. In my view, this manuscript can be accepted for publishing on its current stand.

Thank you for the positive oppinion and the acceptance of the paper in its present form. 

Reviewer #2:

The paper by Marianna et al. analyzed the stability of wheat lines with increased level of arabinoxylan in three years experiment. The author found that the contents of WE-fibre were highly heritable and wheat lines with high contents of WE-fibre also have good and stable processing properties. With no doubt, the research can arouse interests for the readers who engage in breeding wheat cultivars with the good quality and improved health benefits. However, the paper should add more detailed information in order to increase its theoretical component. The manuscript also has some grammar errors and inaccuracies should be revised. Some detailed suggestion for author as below:

1. Please describe the agricultural traits and quality traits of the 31 improved lines and 4 parents

Thank you for this note which was useful to consider. In the manuscript we referred to our previous paper (Tremme-Bede et al. 2017) several times in which we introduced the development of the lines studied in this paper and their most important agronomical and processing quality traits. As we introduced the most important traits of the lines there and do not want to fall into the problem of repetition, finally we decided to keep the original way and logic of the discussion, focusing on the most important traits being in relation to the arabinoxylan content. This is the reason for example that from the quality traits only the water absorption is discussed in more details.

2. The contents of protein and starch are low heritability in many reports. However, this research shows the heritability of the contents of protein and starch is very high (0.851 and 0.828, respectively). Why the results of this research are different from other reports, please discuss more about this.

Thank you. This is also an important note as well and I agree with that. The high effect of the genotype on the protein and starch content in this study might be resulted by fact that the studied lines went through several cycles of selection during the different generations and those lines were selected which not only have high AX, but also have high protein content and/or quality. This information or explanation is now added to the text. 

3. In the methods part, please supply the methods for the test weight and thousand kernel weight.

The standard methods and the instruments used for the measurements were corrected and added.

4. The significant marker in Table1 should be * instead of X, please revise it .

Changes were made.

5. Line 89, there should be a full stop(.) after the ‘respectively’.

It was done so. 

6. Line 347, ‘heritability values for TOT-AX then for WE-AX ‘ should be ‘heritability values for TOT-AX than for WE-AX ‘.

It was done so.

7. Line 349, ‘WE-AX then the G’ should be ‘WE-AX than the G’.

It was done so

Journal Requirements:

- Datas are available in a previous paper: Tremmel-Bede et al. 2017. 

- Figures were checked with PACE. 

- The manuscript was formatted to the PLOSOne style, both the title page and the main body of the paper. 

- The text of the introduction and the methods were compared to the literature and the overlapping text were rephrased. 

- Mentioning Figure 4 was deleted as it does not exist.

- Reference style and order were changed according to PLOSOne rules.

---

## [Decision Letter · Decision Letter 1]

24 Apr 2020

Stability analysis of wheat lines with increased level of arabinoxylan

PONE-D-20-06968R1

Dear Dr. Rakszegi,

We are pleased to inform you that your manuscript has been judged scientifically suitable for publication and will be formally accepted for publication once it complies with all outstanding technical requirements.

With kind regards,

Aimin Zhang, Ph.D.

Academic Editor

PLOS ONE

Additional Editor Comments (optional):

Reviewers' comments:

Reviewer's Responses to Questions

**Comments to the Author**

1. If the authors have adequately addressed your comments raised in a previous round of review and you feel that this manuscript is now acceptable for publication, you may indicate that here to bypass the “Comments to the Author” section, enter your conflict of interest statement in the “Confidential to Editor” section, and submit your "Accept" recommendation.

Reviewer #2: All comments have been addressed

2. Is the manuscript technically sound, and do the data support the conclusions?

Reviewer #2: Yes

3. Has the statistical analysis been performed appropriately and rigorously? 

Reviewer #2: Yes

4. Have the authors made all data underlying the findings in their manuscript fully available?

Reviewer #2: Yes

5. Is the manuscript presented in an intelligible fashion and written in standard English?

Reviewer #2: Yes

6. Review Comments to the Author

Reviewer #2: The authors have clarified several of the questions I raised in my previous review.Everything is fine with the manuscript now.

7. PLOS authors have the option to publish the peer review history of their article (what does this mean?). If published, this will include your full peer review and any attached files.

Reviewer #2: No

---

## [Editor Report · Acceptance letter]

28 Apr 2020

PONE-D-20-06968R1 

Stability analysis of wheat lines with increased level of arabinoxylan 

Dear Dr. Rakszegi:

I am pleased to inform you that your manuscript has been deemed suitable for publication in PLOS ONE. Congratulations! Your manuscript is now with our production department. 

With kind regards,

on behalf of

Prof. Aimin Zhang 

Academic Editor

PLOS ONE